# PEML: Prototype-enhanced Meta-learning for Multi-lingual Text Classification

## Abstract

Multi-lingual text classification is a challenging task in natural language processing, which not only faces language differences between multiple languages but also faces the challenge of scarce annotated data. This paper proposes a prototype-enhanced meta-learning (PEML) method to address the challenges in the multi-lingual text classification task. The PEML method consists of two steps: firstly, to enhance the model's ability to understand multi-lingual samples, we design a multi-lingual label-fusion technique to better map labels from different languages into a unified semantic space; secondly, in response to the problem that class prototypes for support sets are difficult to apply to query sets in meta-learning, we use a query-enhanced technique to associates the prototype vectors of the support set with samples in the query set. After training with our method, the classification model can quickly update the class prototypes to the data distribution of the query set, thereby expanding the model's multi-lingual classification ability from the support set to the unseen query set. Extensive experiments demonstrate that the proposed method significantly outperforms state-of-the-art methods in multi-lingual text classification tasks. The code and data will be released on GitHub.

## 1 Introduction

Multi-lingual text classification (MLTC) tasks have wide applications in practical scenarios such as public opinion monitoring and industry intelligence analysis (Brauwers & Frasincar, 2023; Minaee et al., 2022). Due to language and culture differences, it is not easy to annotate text data in different languages using the same classification criteria since annotators need to understand both language differences and annotation standards (Snell et al., 2017; Bao et al., 2020). Therefore, Current researches mainly focus on improving classification performance with only a few annotated samples (few-shot scenario) (Shliazhko et al., 2024; Liu et al., 2023).

Prototypical Networks (PN) are frequently used for solving few-shot text classification tasks (Snell et al., 2017; Bao et al., 2020; Luo et al., 2021). A prototype vector is considered to be representative of a class, which is constructed with samples in the support set (Lei et al., 2023). Query samples are judged to belong to a class based on the distance to the prototype vector (Liu et al., 2024). However, in MLTC tasks, intra-class language differences increase the computational difficulty of class prototypes, which can easily lead to insufficient representativeness of class prototypes and incorrect classification of query samples (Han et al., 2023; Li et al., 2024a).

Recently, several methods were proposed to address intra-class sample differences in PN research. For instance, MLADA (Han et al., 2021) introduced adversarial domain adaptive networks to reduce intra-class differences; ContrastNet (Chen et al., 2023) amplified inter-class differences through contrastive learning; Meta-SN (Han et al., 2023) improved the discriminative ability of the model by introducing external knowledge of class labels and a task construction strategy driven by hard samples. However, these methods usually focus on monolingual tasks (Li et al., 2024a; Zhang et al., 2022) and address intra-class sample differences by updating the class prototype from a category perspective (Li et al., 2024b; Lv, 2024), they lack a design for updating the prototype vectors in a multi-lingual scenario, i.e., updating category information and language information simultaneously.

To better address the challenges in MLTC tasks, this paper proposes a **P**rototype **E**nhanced **M**eta **L**earning (PEML) method, which introduces a multi-lingual and multi-class fusion and matching

strategy to efficiently learn the class prototype vectors for multi-lingual samples and unseen categories in meta-learning. Specifically, PEML first uses a multi-lingual label-fusion module to better map labels from different languages into a unified semantic space, which enhances the model's ability to understand multi-lingual samples. Then, PEML uses a query-enhanced technique to better associate the prototype vectors of the support set with samples in the query set. In this way, the classification model can quickly update the class prototypes to the data distribution of the query set, thereby expanding the model's multi-lingual classification ability to the unseen query set. The advantages of the PEML method are: 1) it does not require a manually constructed prompt, making it suitable for multi-lingual scenarios; 2) it needs only a few annotated data and does not need external knowledge, making it easily be applied to low-resources languages; 3) extensive experiments show that PEML achieves a new state-of-the-art performance under a multi-lingual few-shot scenario. The main contributions of this paper are as follows:

- We propose a prototype-enhanced meta-learning (PEML) method for MLTC tasks.

- We design a label-fusion module and a query-enhancement module in PEML. The label-fusion module maps samples from different languages of the same category to a unified semantic space, enhancing the model's ability to understand samples in multiple languages. The query-enhancement module associates the class prototype vector with samples in the query set, alleviating the problem of insufficient representativeness of prototype vectors.

- Extensive experiments show that PEML achieves new state-of-the-art performance. We give a detailed analysis of the results. The code and data of this paper will be released on GitHub.

## 2 RELATED WORK

Due to the high cost of manual annotation and the difficulty in unifying multi-lingual labels, few-shot learning has become the mainstream research paradigm in MLTC, which can be categorized into transfer learning and meta learning.

### 2.1 TRANSFER LEARNING

Transfer learning usually transfers knowledge from the source domain to the target domain with multi-lingual pre-trained language models (PLMs). In recent years, the prompt-based method has gradually become the mainstream transfer learning paradigm. This method constructs prompt words to guide the model to stimulate existing knowledge with only a few examples (Liu et al., 2023).

Schick & Schütze (2021) introduced a prompt learning paradigm that requires manual template design. The KPT series method (Hu et al., 2022) guides PLMs to predict the category of query samples by constructing an external knowledge graph. Dong et al. (2023) transformed few-shot text classification into a correlation estimation problem and constructed a universal prompt template. Ji (2024) enhanced text representation and improved text classification performance by introducing entity relationship information from knowledge graphs. Liu & Yang (2024) proposed a knowledge-enhanced prompt learning method (SKPT) that optimizes prompt templates by introducing external knowledge (such as open triplets). Meng et al. (2025) proposed a multi-granularity feature extraction method that integrates semantic relevance information of labels, which improves the performance of the model on label confusion problems. Dementieva et al. (2025) proposed an unsupervised cross-linguistic knowledge transfer method that avoids manual data annotation and utilizes large-scale multi-lingual encoders and translation systems for text classification. Although these methods have achieved improvements, they often require the introduction of external knowledge or manual construction of prompt templates, which increases labor costs and has limited generalization ability when dealing with different types of problems (Gao et al., 2025; Hatefi et al., 2025). In addition, although large language models (LLMs) are currently a research hotspot in natural language processing, there are few works on multi-lingual text classification based on LLMs and prompt learning. **Compared with existing transfer learning methods, the few-shot learning method we proposed does not require the introduction of external knowledge or manually constructed prompt templates and only utilizes the characteristics of the data itself, which has better generalization ability.**

## 2.2 META LEARNING

Meta learning, also known as "learning how to learn", trains a model across many few-shot tasks, each consisting of a support set (for learning task-specific information) and a query set (for evaluating generalization). It can help learning algorithms achieve better generalization for unknown tasks, even in situations where training data is extremely limited or insufficient. The current meta learning methods can be mainly divided into three categories:

1) **Optimization-based methods** focus on "how to optimize". For example, MAML (Finn et al., 2017) and Reptile (Nichol et al., 2018) focus on optimizing the gradient descent process.

2) **Model-based methods** learn an implicit feature space and predict the labels of query samples in an end-to-end manner, but often lack interpretability. Compared with optimization-based methods, model-based methods are usually easier to train, but have weaker generalization ability for out-of-distribution tasks (Hospedales et al., 2020). For example, MANN (Santoro et al., 2016) and SNAIL (Mishra et al., 2018), which aim to build adaptive internal states in fixed modules to achieve fast and efficient parameter updates.

3) **Metric-based methods** learn an appropriate distance or similarity function to distinguish samples from different tasks. When faced with new tasks, this method calculates the distance between input samples and known samples, and then classifies them into the most similar or nearest category, such as Matching Network (Vinyals et al., 2016), Prototypical Network (Snell et al., 2017), and the Relation Network (Sung et al., 2018). Among them, the Prototype Network aggregates all annotated samples in each category into a category prototype, and then classifies them by measuring the distance between the query sample and each category prototype. Lei et al. (2023) proposed the TART method, which converts class prototypes into fixed reference points in a task-adaptive metric space to enhance the model's generalization ability. The LAQDA proposed by Liu et al. (2024) utilizes label information and query samples to optimize class prototypes, to alleviate the problem of large intra-class differences and small inter-class differences between support set samples. Li et al. (2024a) proposed a prototype network optimization method that combines label propagation and attention mechanisms, aiming to improve the quality of prototype representation and metric flexibility. **The PEML method proposed in this paper belongs to the metric-based meta-learning method. Unlike existing methods, the PEML method utilizes a label-fusion and a query-enhancement technique to obtain and update the multi-lingual class prototype vectors, thereby better transferring the classification ability of the model to unseen multi-lingual categories.**

# 3 THE PROPOSED PEML METHOD

## 3.1 PROBLEM FORMULATION

Meta-learning provides an effective paradigm for few-shot text classification, typically adopting the $N$-way $K$-shot task setting. For each task, there are $N$ classes, and each class has only $K$ labeled samples for training. The data in meta-learning is divided into two parts: the source classes $Y_{train}$ and the target classes $Y_{test}$, where $Y_{train} \cap Y_{test} = \emptyset$. In general, meta-learning contains two phases: meta-training and meta-testing.

**Meta-training**: During the meta-training phase, the model is trained on a set of meta-tasks constructed from the training set $D_{train}$, whose class labels are sampled from $Y_{train}$. Each meta-task consists of a support set and a query set. Specifically, for each task, $N$ classes are randomly sampled from $Y_{train}$, $K$ labeled examples are sampled as the support set $S_{train}$ and another $M$ examples as the query set $Q_{train}$ per class, denoted as $S_{train} = \{(x_i, y_i)\}_{i=1}^{N \times K}$ and $Q_{train} = \{(x_j, y_j)\}_{j=1}^{N \times M}$. The model makes predictions about the query set $Q_{train}$ based on the given support set $S_{train}$. Then the model updates the parameters by minimizing the loss with $Q_{train}$.

**Meta-testing**: During the meta-testing phase, the model is used to predict the labels of query samples in $D_{test}$. For each task, $N$ novel classes will be sampled from $Y_{test}$, which is disjoint from $Y_{train}$. Then the support set $S_{test}$ and the query set $Q_{test}$ will be sampled from the $N$ classes, like in meta-training. The query set $Q_{test}$ is denoted as $Q_{test} = \{x_j\}_{j=1}^{N \times M}$, where the label of each example is unknown to the model. The performance of the model will be evaluated through the average classification accuracy on the query set $Q_{test}$ across all the testing tasks.

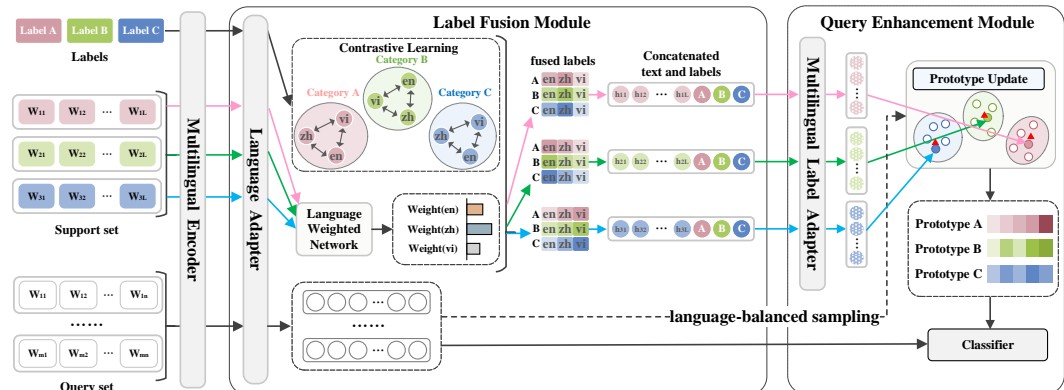

Figure 1: PEML on a 3-way 1-shot classification task. The dashed arrow is the query process.

## 3.2 FRAMEWORK

Figure 1 gives an overview of PEML. First, a *multi-lingual Encoder* gets the representations of the input sentences and label names. **In the label-fusion module**, a *Language Adapter* is applied to enhance language-specific features. Then, contrastive learning is employed to align the semantic spaces of the multiple languages under each label. Meanwhile, a Weighting Network computes language weights for each query sample across multiple languages and adaptively fuses the multi-lingual representations of candidate labels. **In the query-enhancement module**, a *multi-lingual Label Adapter* is used to map the fused representations (label and examples) to a class prototype space. During meta-training, the prototype of the support set is used to calculate losses. During meta-testing, the prototype is used to query samples from the query set $Q_{test}$ as data augmentation to update itself, mitigating the problem of intra-class differences. Finally, query samples are inferred by a *Classifier* which calculates the distance between query samples and the updated prototype vector. Next, we will introduce the PEML method in detail.

## 3.3 LABEL FUSION MODULE

### 3.3.1 INPUT TEXT ENCODING

Firstly, an input text is tokenized and encoded with a multi-lingual encoder (e.g., mBERT) to obtain its representation, i.e., the final-layer hidden state sequence $\{h_1, h_2, \ldots, h_L\} \in \mathbb{R}^{L \times d}$, where $L$ is the sequence length and $d$ is the hidden size. After initial encoding, we design a Language Adapter (LA) module to further optimize the representations of different languages. Different languages will be automatically recognized and input into the corresponding sub-adapter. Each sub-adapter consists of a down-projection of the hidden vector $h_i$ to a lower dimension $r$ (where $r = d/4$), followed by a non-linear transformation, and an up-projection back to the original dimension $d$. The adapted vector is then added to the original vector via a residual connection, followed by layer normalization (LayerNorm). This process is: $\tilde{h}_i = \text{LA}_\Phi(h_i), \quad i = \{1, 2, ..., L\}$. Here, $\Phi$ denotes the trainable parameters of the adapter, and we maintain a separate parameter set for each language, i.e., $\Phi_{en}$, $\Phi_{zh}$, and $\Phi_{vi}$ for English, Chinese, and Vietnamese (we use these three languages for a multi-lingual demonstration in this paper), respectively. This process transforms the original encoding results into a more language-specific representation, enabling the model to capture subtle semantic differences and language-specific features in the multi-lingual embedding space. After applying the adapter transformation to each hidden vector in the sequence, we perform Mean Pooling over the sequence length to obtain a sentence-level vector $\mathbf{v}_{\text{text}} = \frac{1}{L} \sum_{i=1}^{L} \tilde{h}_i, \ \tilde{h}_i \in \mathbb{R}^d$, where $\tilde{h}_i$ denotes the adapter-transformed hidden state at position $i$. This sentence vector preserves the contextual information learned by the multi-lingual encoder through multi-lingual pre-training, while also integrating language-specific features extracted by the adapter.

### 3.3.2 INPUT LABEL ENCODING AND FUSION

In terms of multi-lingual label modeling, assume the current task involves $N$ candidate category labels. For the $j$-th label, its multi-lingual representation includes Chinese, English, and Vietnamese

versions, denoted as $l_j^{(\text{zh})}$, $l_j^{(\text{en})}$, and $l_j^{(\text{vi})}$, respectively. Each version is a sequence of words or subword tokens, represented as $l_j^{(\text{lang})} = [l_1, l_2, \ldots]$, where $\text{lang} \in \{\text{zh}, \text{en}, \text{vi}\}$. We first encode the label names in each language independently using a multi-lingual encoder. The resulting hidden states for each language are mean-pooled and then passed through a language-specific adapter, yielding three language-specific label embeddings: $\hat{l}_j^{(\text{lang})} = \text{LA}_\Phi^{(\text{lang})}\left(\frac{1}{m}\sum_{k=1}^m \text{Enc}(l_j^{(\text{lang})})_k\right)$, where $\text{Enc}(\cdot)$ denotes the multi-lingual encoder, and $\hat{l}_j^{(\text{lang})} \in \mathbb{R}^d$ is the representation of the $j$-th label in language $\text{lang} \in \{\text{zh}, \text{en}, \text{vi}\}$ after processing with the corresponding Language Adapter.

To enable dynamic semantic fusion of multi-lingual label representations, we introduce a language-weighted Network, which takes the pooled text representation $\mathbf{v}_{\text{text}}$ as input and predicts an attention distribution over the three language-specific label embeddings. The Query Network is implemented as a two-layer multilayer perceptron (MLP), whose output is a 3-dimensional vector normalized by a Softmax function to produce the attention weights $(w_{\text{zh}}, w_{\text{en}}, w_{\text{vi}})$. These weights are used to compute a weighted combination of the multi-lingual label embeddings. The final fused multi-lingual label representation for the $j$-th label is given by: $\tilde{l}_j = w_{\text{zh}} \cdot \hat{l}_j^{(\text{zh})} + w_{\text{en}} \cdot \hat{l}_j^{(\text{en})} + w_{\text{vi}} \cdot \hat{l}_j^{(\text{vi})}$.

To further enhance the semantic consistency among multi-lingual label representations, we incorporate a cross-lingual contrastive loss based on InfoNCE (Information Noise Contrastive Estimation) (Gao et al., 2021). For each category label $j$, we obtain its embeddings under three different languages: $\{\hat{l}_j^{(\text{zh})}, \hat{l}_j^{(\text{en})}, \hat{l}_j^{(\text{vi})}\}$. These embeddings are first $\ell_2$-normalized and treated as positive pairs. In contrast, embeddings of different category labels across languages are considered as negative samples. Assume a mini-batch contains $B$ labels in total. The contrastive loss $\mathcal{L}_{\text{contra}}$ is defined as:

$$-\frac{1}{3B}\sum_{j=1}^B \sum_{(a,b)\in\Omega} \log \frac{\exp\left(\text{sim}\left(\tilde{l}_j^{(a)}, \tilde{l}_j^{(b)}\right)/\tau\right)}{\sum_{k=1}^B \exp\left(\text{sim}\left(\tilde{l}_j^{(a)}, \tilde{l}_k^{(b)}\right)/\tau\right)}, \tag{1}$$

where $\Omega = \{(\text{zh}, \text{en}), (\text{zh}, \text{vi}), (\text{en}, \text{vi})\}$, $\text{sim}(\cdot)$ refers to cosine similarity, and $\tau$ is a temperature hyperparameter. This loss encourages the embeddings of the same label across different languages to be semantically aligned, while pushing apart embeddings of different labels. As a result, it improves the cross-lingual alignment of label representations in the shared semantic space.

We concatenate the word vector sequence of the input sentence $[\tilde{h}_1, \tilde{h}_2, \ldots, \tilde{h}_L]$, and the multi-lingual label representations of all $N$ candidate categories in the current task $[\tilde{l}_1, \tilde{l}_2, \ldots, \tilde{l}_N]$ to form the overall input sequence to the multi-lingual label adapter: $[\tilde{h}_1, \tilde{h}_2, \ldots, \tilde{h}_L, \tilde{l}_1, \tilde{l}_2, \ldots, \tilde{l}_N]$.

### 3.4 QUERY ENHANCEMENT MODULE

#### 3.4.1 MULTI-LINGUAL LABEL ADAPTER

To construct a task-adaptive metric space in which intra-class samples are more tightly clustered and inter-class samples are better separated, we introduce a multi-lingual Label Adapter (MLA), which enhances the conventional self-attention framework by explicitly incorporating multi-lingual label semantics into sentence representation learning. The goal of this module is to enhance the model's discriminative capacity across languages by embedding task-specific label information directly into the sentence encoding process.

To enable deep integration between the input sentence and multi-lingual label information, we introduce a set of multi-head attention blocks. This module is parameterized by a set of trainable meta-parameters $\theta$, and is defined as: $\text{MLA}_\theta(Q, K, V) = \sigma(QK^T) \cdot V$, where the pairwise dot-product $QK^T$ measures the similarity amongst features and is used for feature weighting computed through an activation function $\sigma$. Intuitively, each feature of $V$ will get more weight if the dot-product between $Q$ and $K$ is larger. In $\text{MLA}_\theta(Q, K, V)$, following the self-attention mechanism, we have $Q = K = V$. We input $[\tilde{h}_1, \tilde{h}_2, \ldots, \tilde{h}_L, \tilde{l}_1, \tilde{l}_2, \ldots, \tilde{l}_N]$ to the MLA. The output of <CLS> position (denoted as $\tilde{h}_0^*$) serves as the final representation vector $v$ of the sentence: $v = h_0^* = \text{MLA}_\theta\left(\left[\tilde{h}_1, \tilde{h}_2, \ldots, \tilde{h}_L, \tilde{l}_1, \tilde{l}_2, \ldots, \tilde{l}_N\right]\right)$.

By explicitly incorporating multi-lingual label embeddings into the sentence representation construction process, the model not only preserves the original linguistic context of the text but also captures the cross-lingual semantic features of all candidate categories. This mechanism enables the sentence representation to align more closely in the embedding space with the multi-lingual centroid representations of its corresponding labels, thereby significantly enhancing the model's accuracy and robustness in category discrimination. This mechanism demonstrates superior generalization and discriminative capabilities, especially in typical few-shot scenarios where multi-lingual samples are imbalanced or label semantics are ambiguous.

### 3.4.2 PROTOTYPE UPDATE

In multi-lingual few-shot text classification, due to the limited number of samples in the support set $S$ and the semantic variance within classes, the class prototypes constructed solely from the support set often suffer from bias and fail to accurately represent the true class centers. In contrast, the query set $Q$ typically contains more unlabeled samples with a broader distribution. Therefore, we incorporate information from the query set to update and enhance the representational capacity of the class prototypes. For each support sample, we select the top $R$ most similar samples from the query set to assist in constructing more accurate class prototypes. This is based on the intuition that richer sample information can generally lead to better prototype estimation. Meanwhile, to mitigate semantic bias caused by language imbalance, we apply language-balanced sampling to the top $R$ candidate samples. Specifically, we allocate $\lfloor R/3 \rfloor$ samples to each language. Any remaining slots are filled by randomly selecting from the remaining candidate pool, ensuring that the selected samples are as balanced across the three languages as possible.

For an $N$-way $K$-shot classification task, let the $K$ support samples of novel class $c$ be denoted as $\{x_{c1}^S, ..., x_{cK}^S\}$, with their corresponding representations $\{v_{c1}^S, ..., v_{cK}^S\}$. The query set contains $M$ unlabeled samples $\{x_1^Q, ..., x_M^Q\}$, whose representations are $\{v_1^Q, ..., v_M^Q\}$. We treat each sample as a random variable following a Gaussian distribution and use an Optimal Transport (OT) technique that can help align data distributions between query samples and class prototypes from the support set. Specifically, for each sample in the $c$-th class support set $S_c$, we first retrieve its $R$ most similar samples in the query set $Q$ based on the OT distance:

$$M_c^Q = \underset{c \in N}{\operatorname{argmin}} \mathcal{W}(Q, S_c) = \underset{c \in N}{\operatorname{argmin}} \min_{\mathbf{T} \in \mathbf{\Sigma}(Q, x_{ci}^S)} < \mathbf{C}, \mathbf{T} > = \{a_1, a_2, ..., a_R\}, \tag{2}$$

where $\mathbf{C}$ denotes the cost matrix and each element is computed by the Euclidean distance between a query sample and a support sample: $c(x_i^Q, x_j^S) = \| v_i^Q - v_j^S \|_2^2$. $\mathbf{T} \in \mathbf{R}_+^{K \times m}$ represents the optimal transport plan matrix that satisfies the marginal constraints $\{\mathbf{T} \cdot 1_m = Q, \mathbf{T} \cdot 1_K = S_c\}$, indicating the optimal pairing between support and query samples. The optimal transport matrix $\mathbf{T}_c$ can be efficiently solved using the Sinkhorn Algorithm (Cuturi, 2013).

We next adapt the augmented information $M_c^Q$ from the query set $Q$, mapping to the task as follows:

$$\hat{a}_i = \underset{a_i \in M_c^Q}{\operatorname{argmin}} \sum_{j=1}^K \mathbf{T}_c(i, j) \cdot c\left(a_i, v_{cj}^S\right), \ \ i = \{1, ..., R\}, \tag{3}$$

where $\hat{a}_i$ denotes the projected representation of the $i$-th sample representation in $M_c^Q$, and $\mathbf{T}_c(i, j)$ is the element at position $(i, j)$ of the transport matrix $\mathbf{T}_c$. This projection maps the query samples to the distribution space of the support set through the optimal transport matrix Tc to reduce language bias. Previous studies (Liu et al., 2024) have shown that when the cost function is the squared Euclidean norm, this projection process is equivalent to a weighted average of $S_c$ (Courty et al., 2017) as follows: $\hat{S}_c = \operatorname{diag}\left(\mathbf{T}_c 1_{n_c}\right)^{-1} \mathbf{T}_c S_c$, where $\hat{S}_c$ denotes the enhanced support sample representations for class $c$, and $\operatorname{diag}$ is a diagonal matrix. Following previous work (Liu et al., 2024), we obtain the adapted augment information $\hat{S}_c$, which is then combined with the original support sample representations to compute the final prototype of the $c$-th class: $P_c = mean\left(union\left(S_c, \hat{S}_c\right)\right)$. By integrating auxiliary sample information from the query set, this enhanced prototype becomes more robust and better aligned with the true class center, thereby improving classification performance in multi-lingual few-shot scenarios.

Table 1: Experimental results under different few-shot settings.

| Methods | 2-shot | | | 4-shot | | | 8-shot | | |
|---|---|---|---|---|---|---|---|---|---|
| | Precision | Recall | F1 | Precision | Recall | F1 | Precision | Recall | F1 |
| mBERT | 49.31 | 39.15 | 41.10 | 72.83 | 70.46 | 70.60 | 77.21 | 75.31 | 75.37 |
| PET | 28.03 | 23.01 | 25.27 | 31.84 | 26.63 | 27.65 | 41.46 | 40.48 | 39.57 |
| MetricPrompt | 67.49 | 64.55 | 64.45 | 75.09 | 73.29 | 73.13 | 80.90 | 79.79 | 79.80 |
| KPT | 75.48 | 74.16 | 72.18 | 80.22 | 78.84 | 78.67 | 82.50 | 80.78 | 80.81 |
| PBML | 86.77 | 86.41 | 85.87 | 87.57 | 87.39 | 86.95 | 88.89 | 88.65 | 88.76 |
| DML | 11.84 | 14.40 | 10.17 | 29.62 | 23.14 | 22.58 | 52.70 | 43.69 | 44.27 |
| LAQDA | 88.35 | 86.12 | 87.75 | 89.62 | 89.35 | 89.29 | 91.51 | 90.39 | 90.21 |
| EMPT | 88.49 | 88.50 | 88.44 | 89.34 | 89.16 | 89.21 | 90.10 | 89.91 | 90.09 |
| Sailor2-8B | 76.88 | 75.34 | 76.10 | 78.73 | 76.95 | 77.83 | 79.35 | 77.86 | 78.60 |
| PEML(ours) | **91.26** | **90.65** | **90.57** | **92.61** | **92.15** | **92.11** | **93.54** | **93.29** | **93.27** |

### 3.5 Training and Testing Phases

**Training Phase:** For each query sample $x_i^Q$ with representation $v_i^Q$, the model calculates the Euclidean distance between $v_i^Q$ and each class prototype. The resulting distances are passed through a softmax function to estimate the probability that the sample belongs to $c$-th class, as follows:

$$P\left(y_c|x_i^Q, \mathcal{P}\right) = \frac{\exp\left(-||v_i^Q - P_c||_2^2\right)}{\sum_{j=1}^{N} \exp\left(-||v_i^Q - P_j||_2^2\right)}, \tag{4}$$

where $P_c$ denotes the prototype of class $c$, and $N$ is the number of classes in the task. To optimize the model parameters, we use the cross-entropy loss function to compute the prediction error across all query samples and all classes: $\mathcal{L}_{ce} = \sum_{q=1}^{n} \sum_{c=1}^{N} y_{qc} \log P\left(y_c|x_i^Q, \mathcal{P}\right)$, where $y_{qc} = 1$ if the query sample $x_i^Q$ belongs to the $c$-th class; otherwise, $y_{qc} = 0$. $n$ denotes the number of samples in the query set. We define the optimization objective of PEML as: $\mathcal{L} = \mathcal{L}_{ce} + \lambda \mathcal{L}_{contra}$, where $\lambda$ is a hyperparameter that balances the classification loss and contrastive loss. By minimizing $\mathcal{L}$, all the trainable model parameters can be learned.

**Testing Phase:** For a given $N$-way $K$-shot task, we first generate the adapted representations for the query samples and combine them with the original support set representations to construct the final support set. The Prototypical network is then used to predict the class label for each query sample $x_i^Q$, as given by: $\tilde{y} = \arg\max_c P\left(y_c \mid x_i^Q, \mathcal{P}\right)$.

## 4 Experimental Results and Analysis

The experimental settings are introduced in Appendix A. We aim to answer the following questions with the experiments: 1) whether PEML achieves state-of-the-art performance in few-shot MLTC tasks (section 4.1); 2) how does each component contribute to the overall performance (Section 4.2); 3) whether prototype enhancement leads to more compact intra-class distributions and better-separated inter-class representations (Section 4.3); 4) what is the difference when select different number of query samples (Section 4.4); 5) what is the difference with different $\lambda$ (Appendix B).

### 4.1 Main Results

We conducted experiments on multi-lingual text classification tasks under different few-shot conditions. Table 1 shows the experimental results compared with the baseline models.

We can see that the proposed method PEML (Ours) outperforms the baseline models in different few-shot settings and evaluation metrics, demonstrating strong performance advantages. Specifically, under the 2-shot setting, PEML achieved an F1 score of 90.57%, an improvement of 2.82% and 2.13% compared to the strong baselines LAQDA (87.75%) and EMPT (88.44%), respectively.

Table 2: The ablation study results. "PEML w/o LF" removes the Label Fusion module, "PEML w/o LA" removes the Language Adapter, "PEML w/o CL" removes the Contrastive Learning process, "PEML w/o QE" removes the query enhancement module, "PEML w/o MLA" removes the multi-lingual Label Adapter, "PEML w/o PU" removes the Prototype Update process.

| Methods | 2-shot | | | 4-shot | | | 8-shot | | |
|---|---|---|---|---|---|---|---|---|---|
| | Precision | Recall | F1 | Precision | Recall | F1 | Precision | Recall | F1 |
| PEML | **91.26** | **90.65** | **90.57** | **92.61** | **92.15** | **92.11** | **93.54** | **93.29** | **93.27** |
| PEML w/o LF | 82.75 | 81.43 | 82.08 | 84.42 | 83.21 | 83.59 | 86.96 | 85.91 | 86.61 |
| PEML w/o LA | 86.67 | 85.26 | 85.87 | 88.03 | 87.74 | 87.62 | 90.85 | 89.43 | 90.23 |
| PEML w/o CL | 84.22 | 83.43 | 83.73 | 85.98 | 84.85 | 85.41 | 88.08 | 87.84 | 87.96 |
| PEML w/o QE | 85.54 | 83.24 | 84.36 | 87.26 | 87.16 | 87.21 | 90.06 | 89.82 | 89.96 |
| PEML w/o MLA | 91.24 | 90.59 | 90.48 | 91.03 | 90.74 | 90.62 | 92.08 | 91.73 | 91.85 |
| PEML w/o PU | 86.44 | 85.58 | 85.65 | 87.86 | 87.07 | 87.12 | 88.23 | 88.01 | 88.02 |

It also far surpasses traditional fine-tuning and prompt-based methods such as mBERT, PET, KPT, etc. PEML also outperforms the state-of-the-art multi-lingual LLM, such as Sailor2. The performance of Sailor2 is lower than methods based on smaller PLMs (e.g., PBML, LAQDA EMPT). This result indicates that even the state-of-the-art multi-lingual LLM faces a challenge in multi-lingual few-shot text classification tasks. In summary, the 2-shot results verify PEML's ability to construct more accurate category prototypes under multi-lingual and extremely few-shot conditions.

In the 4-shot and 8-shot settings, most baseline models showed steady performance improvements. The DML method has the most significant improvement when given more labeled samples. Taking the F1 metric for instance, DML's 4-shot has increased by more than 2 times compared to 2-shot, and 8-shot has increased by nearly 2 times compared to 4-shot. This indicates that the DML method can learn multi-lingual classification patterns from an increasing number of examples. However, there is still a significant gap between it and the best-performing models. Among all models, PEML still performed the best. The results again verify PEML's good capability for few-shot MLTC tasks.

## 4.2 ABLATION STUDY

To evaluate the impact of the two modules proposed in this paper on model performance, we conducted ablation experiments under different few-shot settings, and the results are shown in Table 2. We can see that **when removing the Label Fusion (LF) modules**, the performance of PEML drops significantly (the F1 metric drops 8.49/8.52/6.66 on the 2/4/8-shot setting, respectively). The results show that the LF module plays an important role in helping the model understand multi-lingual text. When the Language Adapter (LA) was removed, the F1 score decreased by 4.70, 4.49, and 3.04 percentage points under the three settings, respectively. The results indicate that LA helps capture language-specific features and enhances cross-lingual representation. The Contrastive Learning (CL) process also helps to map different languages into a unified semantic space, removing CL causes performance declines. **When removing the Query Enhancement (QE) modules**, the F1 values dropped to 84.36/87.21/89.96 on the 2/4/8-shot setting, respectively. Among them, the performance degradation was most significant under the 2-shot setting, reaching 6.21 percentage points. This indicates that in extremely few-shot scenarios, the PE module can alleviate the problem of sparse or insufficiently representative support set samples, improve the robustness and discrimination of class prototypes. When the multi-lingual Label Adapter (MLA) module was removed, the F1 value decreased by 0.09/1.49/1.42 under the 2/4/8-shot setting, respectively. This shows that the MLA module introduces semantic information of multi-lingual labels, which helps to build a more task-consistent and semantically distinguishable representation space. When the Prototype-Update (PU) module was removed, the F1 value decreased by 4.92/4.99/5.25 under the 2/4/8-shot setting, respectively. This shows that the PU module helps to build more robust prototype vectors, which increases the classification performance.

## 4.3 VISUALIZATION

To evaluate the ability of different models in building better class prototypes, we used t-SNE (Van der Maaten & Hinton, 2008) to visualize the sample representations and class prototypes generated by PBML, LAQDA, and our method. We randomly selected 3 categories with 150 query samples per

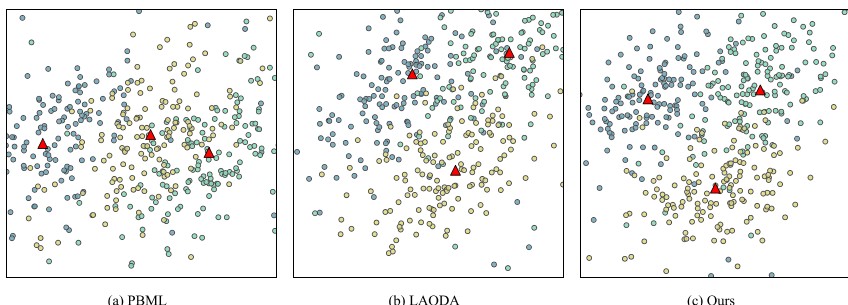

|  (a) PBML | (b) LAQDA | (c) Ours |

Figure 2: Visualization of sample text representations extracted from three new categories on the multi-lingual industry dataset. The triangle represents the calculated category prototype.

Table 3: Experimental results under different $R$.

| Number of $R$ | 2-shot | | | 4-shot | | | 8-shot | | |
|---|---|---|---|---|---|---|---|---|---|
| | Precision | Recall | F1 | Precision | Recall | F1 | Precision | Recall | F1 |
| R=3 | 89.99 | 89.13 | 88.99 | 92.19 | 91.59 | 91.52 | 92.86 | 91.79 | 91.73 |
| R=6 | 91.12 | 90.43 | 90.34 | 92.36 | 92.01 | 91.93 | 93.22 | 92.57 | 92.53 |
| R=12 | **91.26** | **90.65** | **90.57** | **92.61** | **92.15** | **92.11** | **93.54** | **93.29** | **93.27** |
| R=24 | **91.26** | 90.61 | 90.54 | 92.59 | 92.12 | **92.11** | 93.19 | 92.46 | 92.40 |

category from the collected data. The results are shown in Figure 2 (a), (b), and (c). We can see that our PEML method makes samples have stronger compactness within the class, so as to alleviate the problem of insufficient representativeness of support sets in a multi-lingual scenario.

### 4.4 EXPERIMENTS WITH DIFFERENT QUERY SAMPLES

To verify the impact of the number of query samples $R$ in the prototype update module on the model performance, we conducted parameter sensitivity experiments in different few-shot settings (2/4/8-shot), setting $R=\{3,6,12,24\}$, respectively. It means that we introduce 1, 2, 4, and 8 auxiliary samples in each language. The experimental results as shown in Table 3. We can see that in 2/4/8-shot settings, the overall F1 value increases when $R$ increases, indicating that introducing more auxiliary samples appropriately can help improve the representation quality of the prototype. Among them, under the 2-shot setting, the F1 value increased by about 1.55 percentage points (from 88.99% to 90.54%) from $R=3$ to $R=24$; Under the conditions of 4-shot, F1 increased by 0.59% from $R=3$ to $R=24$. This indicates that under few-shot conditions, the gain brought by auxiliary samples is more significant, and additional language-balanced samples can alleviate the problem of class center shift, thereby improving classification performance. However, under the conditions of 8-shot, the performance gain tends to drop when $R$ is larger than 12, indicating that too many auxiliary samples may bring redundant information or noise, affecting the query-enhancement effect. Hence, during the experiments, we select $R=12$.

**For more experiments**, Appendix B shows the experiments for deciding $\lambda$ in the final loss $\mathcal{L}$, Appendix C shows the confusion matrix results of the test set. Appendix D shows experiments on other public MLTC datasets with 6 languages.

### 5 CONCLUSION

We propose a prototype-enhanced meta-learning (PEML) framework for few-shot MLTC tasks. The method introduces two key components: a label-fusion module that jointly encodes label names and text features to construct a task-adaptive metric space, and a query-enhancement module that refines class prototypes via optimal transport using query samples, addressing support set randomness and intra-class variation. Extensive experiments show that PEML achieves state-of-the-art performance. Further experiments are now conducting and will be released in the next version of the paper (including experiments on more languages and different kinds of encoders).

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

Table 4: Statistics of the collected dataset.

| Category | A | B | C | D | E | F | G | H | I |
|---|---|---|---|---|---|---|---|---|---|
| **Chinese** | 20,243 | 3,880 | 8,252 | 18,854 | 7,425 | 9,958 | 9,995 | 15,590 | 19,584 |
| **English** | 14,076 | 1,574 | 2,773 | 8,061 | 3,381 | 4,716 | 5,163 | 9,160 | 7,751 |
| **Vietnamese** | 3,320 | 253 | 210 | 579 | 2,915 | 737 | 2,930 | 2,966 | 2,848 |
| **Total** | 37,639 | 5,707 | 11,235 | 27,494 | 13,721 | 15,411 | 18,088 | 27,716 | 30,183 |

## A  EXPERIMENTAL SETTINGS

### A.1  DATASET

Due to the lack of multi-lingual text classification datasets under a unified classification standard in real-world scenarios, we collected a total of 187,194 news articles from publicly available news websites, including 113,781 in Chinese, 56,655 in English, and 16,758 in Vietnamese. The data statistics are shown in Table 4. The average word length of Chinese/English/Vietnamese news is around 245/326/513, respectively. The collected data covers multiple categories. We use the classification tags provided by the website and classify the data into 9 categories based on the same classification criteria. The Category index A is "Agriculture, forestry, animal husbandry and fishery", B is "Mining", C is "Manufacturing", D is "Transportation, storage and postal services", E is "Finance", F is "Real estate", G is "Education", H is "Health and social work", I is "Culture, sports and entertainment". To verify the ability of few sample learning, data from 9 categories were divided into training category, validation category, and testing category in a 4:2:3 ratio in the experiment. We also tested with a 3:3:3 division, and the results are similar to those with a 4:2:3 division.

### A.2  EVALUATION METRICS

Following previous work (Liu et al., 2024; Lv, 2024), we use Precision, Recall, and F1 score to measure the performance of text classification. The above indicators are calculated based on weighted averages in subsequent sections to ensure that the results accurately reflect the model's overall performance in each category.

### A.3  TRAINING DETAILS

The experimental implementation is based on the PyTorch (Paszke et al., 2019) and Huggingface (Wolf et al., 2020) frameworks. To ensure fairness in comparison, all methods used BERT-base-multi-lingual-uncased (Devlin et al., 2019) as the backbone model, and all experiments were run on an NVIDIA RTX 4090 GPU. The model parameters were optimized using the AdamW optimizer (Loshchilov & Hutter, 2019), with a learning rate set to 1e-6. Early stopping strategy is executed when the performance of the validation set does not improve for 20 consecutive epochs. We conducted few-shot classification experiments under the 2-way 2-shot, 2-way 4-shot, and 2-way 8-shot settings. Following previous work, 100, 100, and 1000 tasks are randomly sampled for training, validation, and testing in each round for each method, respectively. For each category, 30 query samples in each task are randomly selected. We also test with different numbers of query samples, and the results still verify the effectiveness of PEML. Due to the page limit, we only show the 30 query sample experiments.

### A.4  BASELINE MODELS

This paper chooses the following method as the baseline model (including both transfer learning methods and meta learning methods):

**mBERT** (Devlin et al., 2019) uses the output of the <CLS> position as text representations, and maps them to specific categories for classification through linear layers.

**PET** (Schick & Schütze, 2021) uses manual templates to construct prompts by adding prefixes or suffixes to the input text and masking certain markers, converting the given tasks into fill-in-the-blank phrases.

Please classify the given sentence into one of the 9 categories (A. "Agriculture, forestry, animal husbandry and fishery", B. "Mining", C. "Manufacturing", D. "Transportation, storage and postal services", E. "Finance", F. "Real estate", G. "Education", H. "Health and social work", I. "Culture, sports and entertainment".).
**Examples:** Sentence: "国际教育招生规模进一步扩大。", Category: G. Sentence: "Tin ore production increased by 10%.", Category: B. **Question:** Sentence: "Số lượng du học sinh hai nước qua biên giới tăng hàng năm", Category: ?

Figure 3: The 2-shot prompt example for Sailor2 (Including three languages).

Table 5: Experimental results with different $\lambda$. Best results are highlighted in bold.

| Value of $\lambda$ | 2-shot | | | 4-shot | | | 8-shot | | |
|---|---|---|---|---|---|---|---|---|---|
| | Precision | Recall | F1 | Precision | Recall | F1 | Precision | Recall | F1 |
| $\lambda$=0.05 | 91.05 | 90.58 | 90.42 | 92.32 | 91.87 | 91.83 | 92.88 | 92.96 | 92.92 |
| $\lambda$=0.10 | **91.26** | **90.65** | **90.57** | **92.61** | **92.15** | **92.11** | **93.54** | **93.29** | **93.27** |
| $\lambda$=0.15 | 91.19 | 90.52 | 90.51 | 92.42 | 92.02 | 92.05 | 92.98 | 92.76 | 92.81 |
| $\lambda$=0.20 | 91.16 | 90.51 | 90.50 | 92.21 | 91.93 | 92.01 | 92.44 | 92.33 | 92.32 |
| $\lambda$=0.25 | 91.11 | 90.43 | 90.45 | 92.12 | 91.89 | 91.97 | 92.41 | 92.30 | 92.29 |

**MetricPrompt** (Dong et al., 2023) transforms few-shot text classification into a correlation estimation task, using a prompt model as a correlation measure and supervised training of the model through cross-entropy loss.

**KPT** (Hu et al., 2022) expands the search space of label words by introducing external knowledge and uses a PLM to refine the expanded label word space before prediction, aiming to improve and stabilize the performance of prompt tuning.

**PBML** (Zhang et al., 2022) combines a prompt mechanism and meta learning framework for few-shot text classification. with few samples by assigning label words and template learning.

**DML** (Li et al., 2024b) introduces a dual meta learning mechanism to optimize the teacher and student models through pseudo-label correction and feedback supervision, improving the performance of semi-supervised text classification.

**LAQDA** (Liu et al., 2024) uses label information and query samples to optimize class prototypes, to alleviate the problem of large intra-class differences and small inter-class differences between support set samples.

**EMPT** (Lv, 2024) introduces an efficient prompt optimization method in the meta learning framework, which normalizes label and sample information and uses regression to solve closed-form solutions, improving inference speed and classification accuracy, and enhancing stability under a small amount of meta-training data.

**Sailor2-8B** (https://huggingface.co/sailor2) is a state-of-the-art large language model (LLM) that is specially trained for multiple languages, including the collected English, Chinese, and Vietnamese. LLM is well-known as a few-shot learner, making Sailor2 suitable for comparing our method on a few-shot text classification task. The 2-shot prompt for Sailor2 is shown in Figure 3.

# B   EXPERIMENTS WITH DIFFERENT $\lambda$

We set $\lambda$={0.05, 0.1, 0.15, 0.2} in final loss $\mathcal{L}$ to conduct parameter sensitivity experiments with 2/4/8-shot settings. The results are shown in Table 5. We can see that the F1 value is best when $\lambda$=0.1 in all settings. When $\lambda$ became larger or smaller, the model's performance all dropped. Hence, during the experiments, we select $\lambda$=0.1. However, the overall trend of change is relatively small, indicating that contrastive learning can steadily improve the performance of the model.

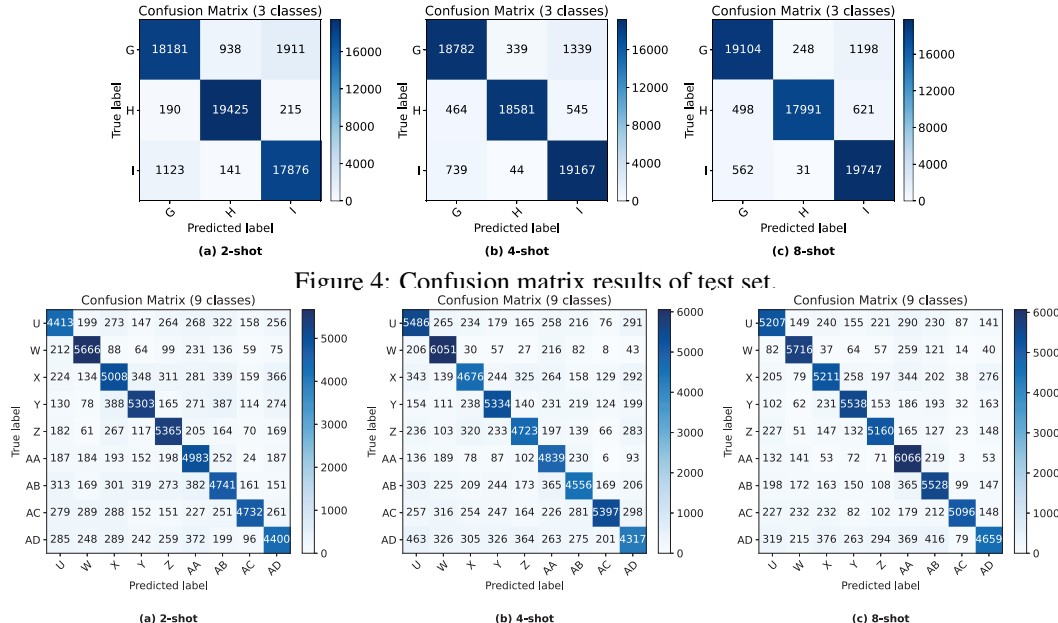

Figure 4: Confusion matrix results of test set.

Figure 5: Confusion matrix results on the test test of the AMR data.

## C    CONFUSION MATRIX RESULTS OF TEST SET

From the confusion matrix shown in Figure 4, it can be observed that as the number of shots increases from 2 to 4 and 8, the overall accuracy gradually improves, multilingual cross-class misclassification continuously decreases, class boundaries become clearer, and intra-class consistency increases. This trend validates the effectiveness of the proposed PEML method: by leveraging multilingual label fusion, labels from different languages are aligned into a unified semantic space, significantly alleviating cross-lingual semantic bias and inter-class confusion. Furthermore, through query enhancement, the model can rapidly adapt class prototypes to the query distribution during inference, thereby overcoming the representation bottleneck of multilingual models in low-resource scenarios.

Table 6: Experimental results with public multilingual datasets.

| Methods | 2-shot | | | 4-shot | | | 8-shot | | |
|---|---|---|---|---|---|---|---|---|---|
| | Precision | Recall | F1 | Precision | Recall | F1 | Precision | Recall | F1 |
| LAQDA | 74.59 | 73.45 | 73.07 | 78.02 | 77.17 | 76.95 | 81.21 | 80.40 | 80.22 |
| PEML | **76.53** | **75.34** | **75.03** | **79.18** | **78.29** | **77.24** | **83.62** | **82.28** | **82.06** |

## D    EXPERIMENTAL RESULTS ON PUBLIC MULTILINGUAL DATASETS

To further validate the effectiveness of our proposed method, we conducted experimental evaluations on a public multilingual dataset. Specifically, we used the *multilingual-amazon-reviews-6-languages* dataset (AMR: Amazon Multilingual Reviews Dataset, available at: https://huggingface.co/datasets/srvmishra832/multilingual-amazon-reviews-6-languages), which covers 6 languages and over 30 categories. We split this dataset into 15 categories for training, 5 for validation, and 9 for testing. Meanwhile, we compared our method with state-of-the-art baseline model LAQDA, and the experimental results along with confusion matrices are presented below.

It can be seen from Table 6 that compared with the baseline model LAQDA, PEML has significant advantages in terms of precision, recall rate and F1 value under 2-shot, 4-shot and 8-shot Settings.

Among them, the F1 value at 2/8-shot outcome LAQDA around 2%. This result demonstrates that PEML shows stronger generalization ability in multilingual classification tasks. From the confusion matrix in Figure 5, it can be seen that the category boundaries of PEML are clear, indicating that it can still maintain good category discriminability in more languages and few-shot scenarios, fully verifying the core design effectiveness of the method in addressing the challenges of language differences and few-shot in MLTC.

