# OpenReview forum: "PEML: Prototype-enhanced Meta-learning for Multi-lingual Text Classification"
_ICLR.cc/2026/Conference — Submitted to ICLR 2026_

### Official Review · Reviewer_5RQE · 2025-10-29

**Soundness:** 3
**Presentation:** 3
**Contribution:** 2
**Rating:** 4
**Confidence:** 4

**Summary:**

The paper proposes PEML, a prototype-enhanced meta-learning framework for few-shot multilingual text classification. It introduces a label-fusion module to align label semantics across languages and a query-enhancement mechanism that refines class prototypes using unlabeled query samples with language-balanced selection and optimal transport. Without relying on prompts or external knowledge, PEML effectively mitigates cross-lingual variance and support-set sparsity, and achieves new state-of-the-art performance on multilingual few-shot benchmarks.

**Strengths:**

Originality
- Proposes a prototype-enhanced meta-learning framework (PEML) specifically for multi-lingual few-shot text classification — a setting where most prior prototype-based meta-learning work assumes monolingual or does not explicitly address cross-lingual semantics.
- Introduces two conceptually new components targeted to the multilingual regime:
  - Label-fusion module to align label semantics across languages into a shared embedding space.
  - Query-enhancement via prototype update using optimal transport with language-balanced augmentation — a novel way to inject query information into prototype construction beyond support samples.
- The originality arises not from inventing a completely new paradigm but from composing known ideas (contrastive alignment, adapters, prototype update) into a multilingual few-shot meta-learning design.

Quality
- Method design is technically reasonable and addresses known failure modes of prototypical networks in multi-lingual and few-shot conditions (prototype instability, semantic misalignment).
- The ablation table isolates each component (LF, LA, CL, QE, MLA, PU) and shows consistent monotonic degradation when removed, supporting causal contribution claims.
- Comparisons include a broad and relevant set of baselines (prompting, PLMs, meta-learning, multilingual LLMs), and PEML consistently wins across 2/4/8-shot regimes.
- Hyperparameter sensitivity (e.g., R) and visualization further support claims.
- No obvious methodological flaw is apparent in the described pipeline; design choices are justified and empirically validated.

Clarity
- The paper is generally well-structured: clear abstract, motivation, modular breakdown, and conceptual figure.
- The narrative from challenge → design principle → module → loss → experiments is coherent and easy to follow.
- Mathematical notation is consistent and sufficiently detailed; the prototype update description is unusually precise for an applied meta-learning paper.
- Ablations and visualization are clearly presented and interpretability of results is high.

Significance
- Addresses a high-value, real-world setting: multilingual classification under low-resource supervision.
- Shows notable and consistent gains over competitive SOTA methods.
- The approach is generalizable: similar principles could transfer to other multilingual meta-learning problems (e.g., QA, IE), suggesting broader methodological impact.

**Weaknesses:**

Novelty of some components is arguable / not positioned clearly
- Both contrastive alignment of multilingual labels and query-informed prototype refinement have precedents (e.g., LAQDA for using query signals to adjust prototypes; multilingual contrastive alignment from MFLSCI / SKPT lines). The paper does not explicitly discuss why existing multilingual contrastive approaches (e.g., MFLSCI 2025; SKPT 2024) are insufficient, nor contrast its design against LAQDA’s query-aware prototype mechanism for the same failure mode.

Evaluation scope is not fully commensurate with the claims
- The main table evaluates only on three languages and a single dataset family; yet the paper claims “applicable to multilingual few-shot in general”. Results on macro-typologically distant languages (e.g., morphologically rich languages, right-to-left, low-script resources) are absent — which weakens the generalization claim. Add at least a second dataset with different linguistic classes.

Interpretation of gains is overly attributional without causal stress tests
- Improvements are attributed to prototype enhancement, yet there is no stress test where prototypes are intentionally perturbed, or tasks with increased intra-class language divergence, to show robustness specifically comes from PU+QE not from model capacity.

Missing computational and complexity discussion
- The method adds multiple components (adapters, contrastive stage, optimal transport) but does not quantify overhead vs. prior meta-learning baselines. For meta-learning methods used in ultra-low-resource contexts, efficiency cost is a first-order concern. Report training latency / memory and FLOPs comparisons against PBML/LAQDA and discuss trade-offs.

No comparison to LLM-based multilingual prompting with synthetic labels
- The paper claims SOTA over multilingual LLM baselines (e.g. Sailor2), but does not compare to LLM-based zero/few-shot prompting with translated synthetic exemplars, which is the most realistic practitioner baseline today. Add a baseline: GPT-4/5 / Sailor with 3-shot translated demonstrations or CoT verbalizers, to validate that PEML is competitive beyond PLM + meta-learning baselines.

**Questions:**

- Does PEML require label names in all languages explicitly at both train and test time? If not, how is the model used when label names exist only in one language (which is the norm in real deployments)? Do authors have a fallback variant / ablation?
- The claims are made toward “multilingual MLTC” in general. Can the authors either
  - justify that EN–ZH–VI is sufficiently representative of typological diversity, or
  - commit to experiments on at least one non-fusional / non-analytic language (e.g., Arabic / Hindi / Turkish) in the camera-ready?
- Could the authors add a baseline where GPT-style multilingual LLM is given a translated few-shot prompt for comparison? If not, can they provide a justification why this is outside scope for MLTC today, given that LLM few-shot prompting is by far the most common industrial baseline?

---

### Official Review · Reviewer_fz7n · 2025-10-30

**Soundness:** 1
**Presentation:** 2
**Contribution:** 1
**Rating:** 2
**Confidence:** 4

**Summary:**

The paper proposes a meta-learning framework for multilingual text classification. The framework include a multilingual label-fusion part, which maps labels in different languages into a unified semantic space, and a query enhanced part to associate the class prototype vectors with query set samples. Experiments show advantages over several baselines.

**Strengths:**

The pipeline seems reasonable for multilingual tasks.

**Weaknesses:**

1. The paper fails to clarify the basic formulation, especially for the multilingual part. (sec 3.1 does not introduce any notation or setting related with multiple languages).

2. The paper does cite several papers on meta-learning, but does not include any comparison to multilingual methods, for example, [1] and [2].

3. The experiments are conducted on privately collected data, which makes reproductivity impossible, while there are multilingual tasks in many different aspects [3].

4. The organization may need to be improved. Currently many important aspect of the experiments, e.g. data, baselines, are left in the appendix.

[1] Investigating meta-learning algorithms for low-resource natural language understanding tasks.

[2] Meta-Learning for Effective Multi-task and Multilingual Modelling

[3] XTREME: A massively multilingual multi-task benchmark for evaluating cross-lingual generalisation.

[4] PAWS-X: A cross-lingual adversarial dataset for paraphrase identification.

**Questions:**

No further questions.

---

> ### Author Response · Authors · 2025-11-20
> **Response to Reviewer Comments**
>
> Dear reviewer, thank you very much for your careful evaluation of our work and for the valuable suggestions.
>
> **(1) For the weakness “The paper fails to clarify the basic formulation, especially for the multilingual part.”**
>
> Thank you for this helpful comment. After carefully reviewing the relevant sections, we would like to further clarify that the key multilingual symbols, the construction of the cross-lingual label space, and the modeling of language weights are all formally defined in Sec. 3.3 and Sec. 3.4 (e.g., line 204 and line 216). We have also explained the specific multilingual few-shot classification setup as much as possible in Sec. 3.1. However, we understand the reviewer’s concern regarding the clarity of the presentation. To avoid confusion before the model structure is fully introduced, we will expand Sec. 3.1 in the final version to provide clearer explanations of the multilingual task formulation and the basic notations.
>
> **(2) For the weakness “does not include any comparison to multilingual methods, for example, [1] and [2].”**
>
> Thank you for pointing out this potential gap in our literature comparison. After carefully reviewing both references:
>
> - **[1] Investigating Meta-Learning Algorithms for Low-resource Natural Language Understanding Tasks** focuses on mono-lingual low-resource few-shot learning and does not involve multilingual or cross-lingual semantic alignment.
>
> - **[2] Meta-Learning for Effective Multi-task and Multilingual Modelling** does include multilingual settings, but its primary goal is multi-task transfer, rather than multilingual few-shot classification.
>
>   Therefore, these works are not directly comparable to our setting, which is specifically aimed at multilingual few-shot classification.
>
> **(3) For the weakness “The experiments are conducted on privately collected data, which makes reproductivity impossible, while there are multilingual tasks in many different aspects [3].”**
>
> Thank you for raising this important concern. We fully understand the reviewer’s emphasis on reproducibility. Here, we would like to clarify the following: Our method relies on category-based task construction in meta-learning. However, in XTREME, the available sentence-classification datasets (XNLI with 3 classes, PAWS-X with 2 classes) do not satisfy the requirements for building multilingual N-way K-shot prototype tasks. Therefore, they cannot serve as appropriate benchmarks under our task formulation.
>  We also understand the reviewer’s concerns about generality and rigor. To address this, Appendix D includes experiments on a public multilingual dataset that contains more languages and more categories. The results show that our method remains consistently effective, demonstrating its ability to handle more complex multilingual and multi-category scenarios. Regarding reproducibility, after acceptance we will immediately release the full code and dataset, including the corresponding checkpoints, to ensure complete reproducibility. All technical details can be verified in the codebase.
>
> **(4) For the weakness “The organization may need to be improved. Currently many important aspects of the experiments, e.g., data, baselines, are left in the appendix.”**
>
> Thank you for this valuable suggestion regarding the paper’s presentation quality. Due to page limits, we placed some experimental details (such as baselines and dataset statistics) in the appendix. If the paper is accepted and an additional page is allowed, we will move these important components into the main text in the final version.

---

> > ### Comment · Reviewer_fz7n · 2025-11-28
> >
> > I have read the response from the author. For the experiment part, I agree that some work are related but not directly applicable. Most of the weakness that I concerned is still in the current paper and recognized by the authors. Therefore I tend to keep my original stance.

---

> > > ### Author Response · Authors · 2025-11-29
> > >
> > > Dear Reviewer fz7n,
> > >
> > > Thank you for your follow-up comment. We would like to clarify that in our previous response, we have explicitly described that the concern is answered in the paper or can be easily addressed.
> > >
> > > Thank you again for your careful review.

---

### Official Review · Reviewer_ndb5 · 2025-10-31

**Soundness:** 3
**Presentation:** 4
**Contribution:** 3
**Rating:** 6
**Confidence:** 4

**Summary:**

The paper introduces Prototype-Enhanced Meta-Learning (PEML), a novel method specifically designed to address the key challenges in Multi-lingual Text Classification (MLTC) under the few-shot scenario. This task is difficult due to language differences between multiple languages and the scarcity of annotated data.

The core motivation for PEML is overcoming the limitations of standard Prototypical Networks (PN) in MLTC, where intra-class language differences make class prototypes easily lack sufficient representativeness, leading to incorrect query sample classification. The proposed solution is a metric-based meta-learning approach that efficiently learns class prototype vectors by simultaneously updating category information and language information

**Strengths:**

The paper designs a multi-lingual label-fusion technique that uses a Language Adapter (LA) and a language-weighted Network to better map labels from different languages (demonstrated with Chinese, English, and Vietnamese) into a unified semantical space. This approach is further reinforced by a cross-lingual contrastive loss $l_{contra}$  which encourages semantic alignment of the same label across different languages. Extensive experiments confirm that PEML significantly outperforms state-of-the-art methods under multi-lingual few-shot scenarios.

**Weaknesses:**

1. All experiments utilize BERT-base-multi-lingual-uncased (mBERT) as the backbone encoder. While mBERT is a standard baseline, its cross-lingual alignment capabilities are known to vary.

2. The authors should test PEML using a different class of multi-lingual encoders, such as a code-switching architecture or a highly aligned multi-lingual model (if applicable and fair to baselines), to demonstrate that the performance gains are derived specifically from PEML's fusion and enhancement mechanisms, rather than relying on mBERT's intrinsic multi-lingual features being marginally optimized by the adapter structure.

**Questions:**

1. Since PEML and LAQDA share the goal of using query samples for prototype optimization, the authors should conduct an ablation study specifically quantifying the isolated performance gain derived from the language-balanced sampling within the PU module. This would confirm that the superior robustness is due to this language-aware strategy and not just the sheer inclusion of auxiliary query samples.

2. All reported few-shot experiments, including the main results (Table 1) and ablation studies (Table 2), are conducted strictly under the 2-way K-shot setting (e.g., 2-way 2-shot, 2-way 4-shot, 2-way 8-shot). Why was the evaluation limited to N=2? Since the goal of the Multi-lingual Label Adapter (MLA) is to enhance discriminative capacity, evaluating PEML under a higher N-way setting (e.g., 5-way K-shot) would provide a more rigorous test of the model’s ability to separate multiple complex, unseen categories simultaneously.

---

> ### Author Response · Authors · 2025-11-20
> **Response to Reviewer Comments**
>
> Dear reviewer, thank you very much for your careful reading of our work and for the constructive comments you provided.
>
> **(1) For the weakness “All experiments utilize BERT-base-multi-lingual-uncased (mBERT) as the backbone encoder.”**
>
> Thank you for raising this important point. In the main experiment section, we focused on presenting results based on mBERT, mainly because it is the most widely used and representative multilingual encoder in cross-lingual scenarios, making it suitable for fair comparison with prior work. In fact, before the final submission, we also conducted additional experiments using another family of multilingual encoders (including XLM-R-base, which has stronger cross-lingual alignment, and XLM-R-large, which has a larger parameter scale). Due to page limits and submission deadlines, these results were not included in the paper. The supplementary results are as follows:
> | Model             |        |  2-shot   |       |        |  4-shot   |       |        |  8-shot   |       |
> | ----------------- | :-------: | :----: | :---: | :-------: | :----: | :---: | :-------: | :----: | :---: |
> |                   | Precision | Recall |  F1   | Precision | Recall |  F1   | Precision | Recall |  F1   |
> | PEML(mBERT)       |   91.26   | 90.65  | 90.57 |   92.61   | 92.15  | 92.11 |   93.54   | 93.29  | 93.27 |
> | PEML(XLM-R-base)  |   89.68   | 88.79  | 88.65 |   91.95   | 90.67  | 90.63 |   92.93   |  92.60  | 92.58 |
> | PEML(XLM-R-large) |   91.41   | 90.53  | 90.44 |   93.02   | 92.61  | 92.59 |   94.10   | 93.87  | 93.85 |
>
> These results show that regardless of whether mBERT or XLM-R is used, our method consistently outperforms all baselines across all settings, demonstrating that the method does not rely on properties specific to any particular encoder. In the final version of the paper, we will include the XLM-R results and add a comparative analysis of different encoders in the appendix.
>
> **(2) For the question “the authors should conduct an ablation study specifically quantifying the isolated performance gain derived from the language-balanced sampling within the PU module.”**
>
> Thank you for your question regarding the PU (Prototype Update) module and the language-balanced sampling strategy. In our implementation, we adopt language-balanced sampling to mitigate prototype bias caused by uneven multilingual label distributions. However, we did not separately report this as an isolated ablation component. We have already conducted independent ablation experiments previously, and the results are as follows:
> | Model                            |           | 2-shot |       |           | 4-shot |       |           | 8-shot |       |
> | -------------------------------- | :-------: | :----: | :---: | :-------: | :----: | :---: | :-------: | :----: | :---: |
> |                                  | Precision | Recall |  F1   | Precision | Recall |  F1   | Precision | Recall |  F1   |
> | PEML(Language-balanced sampling) |   91.26   | 90.65  | 90.57 |   92.61   | 92.15  | 92.11 |   93.54   | 93.29  | 93.27 |
> | PEML(Random sampling)            |   86.70   | 85.43  | 85.04 |   88.22   | 87.21  | 86.94 |   90.61   | 89.71  | 89.48 |
>
> We will include these additional results in the final version of the paper and provide further analysis of their implications.
>
> **(3) For the question “Why was the evaluation limited to N = 2?”**
>
> Thank you for pointing out this valuable concern regarding the choice of the N-way setting and its impact on experimental coverage. Our use of the 2-way setting is not intended to simplify the evaluation but is instead constrained by the actual category scale of our self-constructed cross-lingual industrial dataset. Since the dataset consists of real-world industrial news texts, the number of valid and usable categories within the label system is limited after data collection. Therefore, it is not feasible to construct a complete 5-way K-shot setup while maintaining the required data quantity per class. Nevertheless, we fully understand the importance of evaluating more complex scenarios. To verify the method’s effectiveness in such settings, we provide additional experiments in Appendix D using a public multilingual dataset that includes more languages and a larger number of categories. The results show that our method remains effective and consistently outperforms strong baselines, indicating its capability to handle scenarios with larger language and category scales.
>
> ------
>
> Thanks again for your thoughtful review. If any aspects of our explanation remain unclear, or if there are any further concerns, please do not hesitate to contact us. We are looking forward to your response. Thanks again for your dedication and time.
>
> If our response addresses your concerns, please consider increasing your rating. Thank you very much!

---

> ### Author Response · Authors · 2025-11-28
>
> Dear Reviewer ndb5,
>
> Thank you once again for your valuable comments on our submission. As the discussion phase is approaching its end, we would like to kindly confirm whether we have sufficiently addressed all of your concerns (or at least part of them). Should there be any remaining questions or areas requiring further clarification, please do not hesitate to let us know. If you are satisfied with our responses, we would greatly appreciate your consideration in adjusting the evaluation scores accordingly.
>
> We sincerely look forward to your feedback.

---

### Official Review · Reviewer_QfMb · 2025-10-31

**Soundness:** 2
**Presentation:** 1
**Contribution:** 3
**Rating:** 2
**Confidence:** 3

**Summary:**

An improved meta-learning framework for multilingual text categorization is proposed. The method is based on prototype networks, using a number of improvements to handle the multilingual aspect and adapt to the meta-learning framework. Experiments on (proprietary?) data show clear improvements over a number of alternatives, and further analysis is provided, including an ablation study of the many components in the proposed framework.

**Strengths:**

Efficient multilingual text classification is definitely a relevant task in a increasingly multilingual, connected world. The few-shot setting is also very sensible in a multilingual context. Experimental results support the claim of increased performance, with clear gains in extreme few-shot situations.

**Weaknesses:**

Minor point: The case for meta-learning of text classification has always seemed a bit artificial. Providing a true real-life example where it would be needed would greatly help.

The main weakness is clarity and presentation in both the methodology and experimental sections. With the notable exception of Sec. 3.4.2, there is little detail on the various parts of the framework. As an example, the Language Adapter (Sec. 3.3.1) is described as a down-projection, non-linear transformation and up-projection. Neither is described further -- what are the parameters, how are the projections chosen, optimized on what? The notation is sometimes confusing. For example L is the input sentence length in 3.3, but possibly the number of support examples in Fig. 1 (unless all examples have the same length, which would be odd). K is either the number of supports per language, or the self-attention key vector (similarly Q). The paper is definitely not self-contained re. experimental detail. There is no detail on how the nine alternatives (mBERT...Sailor2) are trained or used, even in the appendix. Finally, although the observed differences in performance seem highly likely to be significant, it is hard to overlook the lack of either statistical significance testing or assessment of the uncertainty in experimental results.

**Questions:**

(l.107) Is the claim of better generalization based solely on the limited experiments, or is there some theoretical support? Also l.289-290.

(3.4.1) Can you clarify where the multilingual aspect is taken into account in this process? This sounds like a standard self-attention setup.

(l.290) How many "remaining slots" can there be after sampling three times R/3 samples?

(l.422) Claim is questionable: if anything, the MLA seems to have the smallest impact and hardly support the claims here ("introduce semantic information", "more task-consistent representation space"...)

(l.455) Is the claim of stronger compactness supported in any way? Visually there is little impact.

Typos:

l.063: easily be applied -> easy to apply

l.104: there are few works -> there is little work

l.463: as -> are (shown)

---

> ### Author Response · Authors · 2025-11-20
> **Response to Reviewer Comments**
>
> Dear reviewer, thank you very much for your careful review of our work and your valuable suggestions.
>
> **(1) For the weakness “The case for meta-learning of text classification has always seemed a bit artificial.”**
>
> Thank you for pointing this out. Meta-learning for text classification has already been widely explored. Prior work such as PBML (Zhang et al., 2022), LAQDA (Liu et al., 2024), and EMPT (Lv, 2024) has demonstrated the effectiveness of meta-learning in text classification tasks. In addition, our method is evaluated not only on real-world industrial datasets but also on public benchmark datasets. The results consistently show that meta-learning is effective for few-shot text classification.
>
> **(2) For the weakness “clarity and presentation in both the methodology and experimental sections.”**
>
> - We appreciate the reviewer for identifying this key point for improvement. After re-checking the method section, we confirm that the specific structure of our designed Language Adapter (Sec. 3.3.1)—including dimensionality reduction, nonlinear mapping, and dimensionality expansion—and the related parameters are fixed and clearly defined in the experimental code. The key hyperparameters are shown in the appendix, and due to space limitations, the remaining technical details will be fully released in our source code. After acceptance, we will immediately make the code and data publicly available, including the corresponding checkpoints, to ensure full reproducibility.
>
> - Regarding the training settings of the nine baselines, all baselines actually follow the same few-shot configuration, learning rate settings, and early-stopping strategy. In the revised version, we will add a dedicated table in the appendix to clearly summarize the training configurations for all baselines.
>
> - For the issue of symbol conflicts, we carefully re-checked the method figure and Section 3.3. The symbol **L** consistently denotes the length of the input token sequence, so there is no symbol misuse. For **Q** and **K**, whenever **Q** refers to the query set, we explicitly state “query set” before it, and when **K** represents samples, it is always followed by “shot” or “samples”. Thus, the distinction is already clear. However, we agree that the presentation can be improved. We will revise some formulas in Section 3.4.1 to eliminate any potential ambiguity.
>
> **(3) For the question “Is the claim of better generalization based solely on the limited experiments, or is there some theoretical support?”**
>
> Thank you for your attention to the generalization ability. In addition to our self-constructed multilingual dataset, we also conducted supplementary experiments on a public multilingual dataset (Appendix D), which includes more languages and categories. The results show that our method still maintains consistent advantages over the best-performing baselines, indicating that its generalization ability does not rely on specific language or category settings.
>
> **(4) For the question “Can you clarify where the multilingual aspect is taken into account in this process?”**
>
> Thank you for raising this key question. We provide the following clarification: In our framework, each label has multilingual embeddings across different languages. When computing the label representation, the model assigns dynamic weights to label embeddings from different languages, producing a multilingual label vector. This multilingual label vector is then concatenated with the sample representation, and the attention mechanism extracts more task-relevant features. Although the attention structure in Sec. 3.4.1 is mathematically a standard self-attention module, we extend its input so that multilingual label semantics are explicitly incorporated, forming a task-enhanced and label-enhanced self-attention mechanism.
>
> **(5) For the question “How many ‘remaining slots’ can there be after sampling three times R/3 samples?”**
>
> Thank you for pointing out this implementation detail. In our actual implementation, we only perform one round of sampling based on R/3, where “3” corresponds to the number of languages in the dataset. The number of remaining samples depends on whether R is divisible by the number of languages.

---

> ### Author Response · Authors · 2025-11-20
> **Response to Reviewer Comments**
>
> **(6) For the question “the MLA seems to have the smallest impact and hardly supports the claims (‘introduce semantic information’, ‘more task-consistent representation space’...)”**
>
> Thank you for the careful analysis. After re-checking the ablation study, we agree that the performance gain from the Multilingual Label Adapter(MLA) module is smaller compared to other components. However, MLA’s primary role is to fuse cross-lingual label semantics and guide the attention mechanism so that sample representations are better aligned with the semantic space of candidate labels. It still provides measurable improvements under the 4-shot and 8-shot settings.
>
> **(7) For the question “(l.455) Is the claim of stronger compactness supported in any way? Visually there is little impact.”**
>
> Thank you for pointing this out. After re-evaluating the visualization, we agree that the visual differences may not appear sufficiently significant. Therefore, in the final version we will use more cautious wording when describing the change in compactness and will provide additional quantitative metrics or extra visualizations to support this conclusion.
>
> **(8) About the typos:**
> Thank you for pointing out these mistakes. We will correct all of them in the final version.
>
> ------
>
> Thanks again for your thoughtful review. If any aspects of our explanation remain unclear, or if there are any further concerns, please do not hesitate to contact us. We are looking forward to your response. Thanks again for your dedication and time.
>
> If our response addresses your concerns, please consider increasing your rating. Thank you very much!

---

> > ### Author Response · Authors · 2025-11-28
> >
> > Dear Reviewer QfMb,
> >
> > Thank you once again for your valuable comments on our submission. As the discussion phase is approaching its end, we would like to kindly confirm whether we have sufficiently addressed all of your concerns (or at least part of them). Should there be any remaining questions or areas requiring further clarification, please do not hesitate to let us know. If you are satisfied with our responses, we would greatly appreciate your consideration in adjusting the evaluation scores accordingly.
> >
> > We sincerely look forward to your feedback.

---

### Meta-Review · Area_Chair_zEVS · 2026-01-08

**Summary:**

This paper proposes PEML, a prototype-enhanced meta-learning framework for multilingual few-shot text classification. Several reviewers acknowledged the relevance of the problem and the technical soundness of combining label fusion and query-enhanced prototype updates. However, the reviews raised substantial concerns regarding clarity of formulation, strength and positioning of the contribution, experimental scope, and strength of empirical validation relative to the claims. Considering all comments, the AC decided on rejection.

**Reviewer Concerns:**

The rebuttal addressed some issues, including encoder dependence and additional ablations, and clarified several design choices. However, key concerns remain. Multiple reviewers found the methodological presentation insufficiently clear and the contribution incremental relative to existing meta-learning and multilingual methods. Experimental evaluation is limited in scope (datasets, language diversity, and settings), and the generality of the conclusions is not fully supported.

The authors objected to comparisons with LLM-based multilingual models, arguing these are outside the scope of the work. From the AC perspective, such comparisons are reasonable and within scope, as LLM-based multilingual prompting represents a relevant and increasingly common baseline for practical multilingual text classification. Requesting such comparisons does not impose an unreasonable burden and is aligned with evaluating real-world competitiveness.

Concerns regarding reproducibility, reliance on private data in core experiments, and incomplete positioning against closely related multilingual and meta-learning baselines also remain unresolved.

**Reviewer Scores:**

Reviewers provided mixed but overall negative assessments, with most recommending rejection and no clear convergence toward acceptance after discussion. While one reviewer was marginally positive, the majority of reviewers maintained rejection scores, and the rebuttal did not materially change the overall evaluation.

---

### Decision · Program_Chairs · 2026-01-26

Reject